# Reduced Expression of a Novel Midgut Trypsin Gene Involved in Protoxin Activation Correlates with Cry1Ac Resistance in a Laboratory-Selected Strain of *Plutella xylostella* (L.)

**DOI:** 10.3390/toxins12020076

**Published:** 2020-01-23

**Authors:** Lijun Gong, Shi Kang, Junlei Zhou, Dan Sun, Le Guo, Jianying Qin, Liuhong Zhu, Yang Bai, Fan Ye, Mazarin Akami, Qingjun Wu, Shaoli Wang, Baoyun Xu, Zhongxia Yang, Alejandra Bravo, Mario Soberón, Zhaojiang Guo, Lizhang Wen, Youjun Zhang

**Affiliations:** 1College of Plant Protection, Hunan Agricultural University, Changsha 410125, China; gonglijun025@163.com (L.G.); guole930323@163.com (L.G.); yefan9605@163.com (F.Y.); yzxmichelle@aliyun.com (Z.Y.); 2Department of Plant Protection, Institute of Vegetables and Flowers, Chinese Academy of Agricultural Sciences, Beijing 100081, China; kangshi0718@163.com (S.K.); zhoujunlei2006@126.com (J.Z.); sun13804560684@163.com (D.S.); qinjianying0203@163.com (J.Q.); liuhongzhu1992@126.com (L.Z.); baiyang15765533722@163.com (Y.B.); makami1987@gmail.com (M.A.); wuqingjun@caas.cn (Q.W.); wangshaoli@caas.cn (S.W.); xubaoyun@caas.cn (B.X.); 3Departamento de Microbiología Molecular, Instituto de Biotecnología, Universidad Nacional Autónoma de México, Apdo. Postal 510-3, Cuernavaca, Morelos 62250, Mexico; bravo@ibt.unam.mx (A.B.); mario@ibt.unam.mx (M.S.)

**Keywords:** *Bacillus thuringiensis*, *Plutella xylostella*, Cry1Ac resistance, trypsin-like midgut protease, protoxin activation

## Abstract

*Bacillus thuringiensis* (Bt) produce diverse insecticidal proteins to kill insect pests. Nevertheless, evolution of resistance to Bt toxins hampers the sustainable use of this technology. Previously, we identified down-regulation of a trypsin-like serine protease gene *PxTryp_SPc1* in the midgut transcriptome and RNA-Seq data of a laboratory-selected Cry1Ac-resistant *Plutella xylostella* strain, SZ-R. We show here that reduced *PxTryp_SPc1* expression significantly reduced caseinolytic and trypsin protease activities affecting Cry1Ac protoxin activation, thereby conferring higher resistance to Cry1Ac protoxin than activated toxin in SZ-R strain. Herein, the full-length cDNA sequence of *PxTryp_SPc1* gene was cloned, and we found that it was mainly expressed in midgut tissue in all larval instars. Subsequently, we confirmed that the *PxTryp_SPc1* gene was significantly decreased in SZ-R larval midgut and was further reduced when selected with high dose of Cry1Ac protoxin. Moreover, down-regulation of the *PxTryp_SPc1* gene was genetically linked to resistance to Cry1Ac in the SZ-R strain. Finally, RNAi-mediated silencing of *PxTryp_SPc1* gene expression decreased larval susceptibility to Cry1Ac protoxin in the susceptible DBM1Ac-S strain, supporting that low expression of *PxTryp_SPc1* gene is involved in Cry1Ac resistance in *P. xylostella*. These findings contribute to understanding the role of midgut proteases in the mechanisms underlying insect resistance to Bt toxins.

## 1. Introduction

*Bacillus thuringiensis* (Bt) are gram-positive entomopathogenic bacteria most widely used as a biopesticide worldwide, and transgenic crops expressing insecticidal toxins produced by these bacteria (transgenic Bt crops) have been planted in 104 million hectares globally in 2018, which has a central role in pest control and global food security [1,2]. However, field-evolved resistance to Bt crops soared from three cases in 2005 to 16 in 2016, documenting an accelerated evolution of practical resistance [3]. Due to the commercial application of Bt proteins, such as Cry proteins for the control of insect pests, it is necessary to probe the resistance mechanism to Cry proteins in order to propose effective strategies to delay the resistance evolution.

Bt Cry proteins are produced as inactive and insoluble crystals formed by protoxins [4]. Cry protoxins are solubilized in the alkaline environment of the midgut and are further processed into activated toxins by midgut proteases when ingested by susceptible larvae [5]. Activated toxins then interact with specific midgut receptor proteins, such as aminopeptidase N (APN), alkaline phosphatase (ALP), cadherin (CAD) and ABC transporters, located in the brush border membrane (BBM) of the midgut epithelium cells from the larvae [6,7] Receptor binding leads to the formation of lytic pores in the membrane that burst cells and finally kill the insects [8]. However, it has been shown that protoxins also bind to specific receptors, and then they are activated by midgut proteases inducing also toxin oligomerization and pore-formation [8,9]. Whether protoxins are activated before or after receptor binding is an important step, transforming the 130 kDa protoxin into a 55–65 kDa activated toxin [4,7]. Trypsin proteases are important midgut proteinases, which participate in Bt Cry protein degradation and protoxin activation [10,11]. It has been reported that alteration of the midgut trypsin genes or trypsin proteolytic activities are linked to Bt resistance in *Plodia interpunctella* (Hübner) (Lepidoptera: Pyralidae) [10], *Ostrinia nubilalis* (Hübner) (Lepidoptera: Pyralidae) [12,13,14], *Spodoptera frugiperda* (JE Smith) (Lepidoptera: Noctuidae) [15], *Helicoverpa armigera* (Hübner) (Lepidoptera: Noctuidae) [16,17], *Aedes aegypti* (L.) (Diptera: Culicidae) [18], *Mythimna unipuncta* (Haworth) (Lepidoptera: Noctuidae) [19], and *Helicoverpa zea* (Boddie) (Lepidoptera: Noctuidae) [20].

The diamondback moth, *Plutella xylostella* (L.), is a cosmopolitan insect pest of cruciferous crops that was the first example of resistance to Bt sprays in the field [21]. The economic damage produced by *P. xylostella* was estimated to be up to USD 5 billion every year [22]. Moreover, since *P. xylostella* was the first documented insect developing field-evolved Bt resistance, it is a good model to understand insect resistance mechanisms to Bt toxins. Previous studies showed that in *P. xylostella*, resistance to the Cry1Ac toxin was not associated with alterations of the *PxABCH1* and *PxCAD* genes [23,24,25]. In contrast, Cry1Ac resistance rather correlated with a mutation in the *PxABCC2* gene [26] or with the differential expression of the *PxmALP*, *PxABCB1*, *PxABCC1*, *PxABCC2*, *PxABCC3,* and *PxABCG1* genes, which were shown to be down-regulated by an enhanced MAPK signaling pathway [27,28,29]. In the MAPK-mediated *trans*-regulatory mechanism, we reported that over-expression of the *PxMAP4K4* gene resulted in down-regulation of diverse midgut genes, thereby conferring a Cry1Ac resistance phenotype. Different *P. xylostella* strains that are resistant to Cry1Ac showed different induction levels of *PxMAP4K4*, resulting in different resistance levels. For example, the SZ-R resistant strain showed moderate resistance levels (662-fold to Cry1Ac) in contrast with the near-isogenic strain (NIL-R) that was highly resistant to Cry1Ac (>3900-fold). The relative expression of *PxMAP4K4* in SZ-R is slightly higher than that of the susceptible DBM1Ac-S strain; thus, expression of some Cry toxin receptors are also down-regulated in the SZ-R strain although at a lower level than that of the NIL-R, which showed the highest constitutive expression of the *PxMAP4K4* gene [27]. In particular, the high Cry1Ac resistance levels in the NIL-R strain does not involve the Cry1Ac protoxin activation mechanism [30]. However, the relationship between the protoxin activation mechanisms in other resistant strains of *P. xylostella* remains unclear.

Here, we compared data from midgut transcriptome and RNA-Seq analyses that were previously done, showing a significant decrease expression of a novel trypsin-like protease in a *P. xylostella* strain SZ-R that shows resistance to Cry1Ac toxin [31,32]. Then, we cloned and characterized the midgut trypsin protease gene of *P. xylostella* (*PxTryp_SPc1*). Finally, we demonstrated that down-regulation of the *PxTryp_SPc1* gene in the midgut tissue of SZ-R strain is related to Cry1Ac resistance by using different genomic, molecular, biochemical, and genetic tools. The conclusions of this study provide a new insight into the Bt resistance mechanism that could give hints for the control of insect pests.

## 2. Results

### 2.1. Comparison of Midgut Protease Activities and Cry1Ac Protoxin Activation between Susceptible DBM1Ac-S and Resistant SZ-R Strains

Previously, differential expression of midgut trypsin-like serine protease (*Tryp_SP*) genes was identified in the Cry1Ac-resistant strain SZ-R in contrast to the susceptible *P. xylostella* strain DBM1Ac-S [31,32]. To determine whether the potential altered *PxTryp_SP* gene expression can change the midgut protease activities and affect Cry1Ac protoxin activation in SZ-R strain, we first compared the midgut protease activities and Cry1Ac protoxin activation in both susceptible DBM1Ac-S and resistant SZ-R strains. The resistant SZ-R strain showed significantly lower caseinolytic protease activity in the midgut extracts than the susceptible DBM1Ac-S larvae (*p* < 0.05; Duncan’s tests; n = 3), likewise, the trypsin activity of resistant SZ-R was also significantly lower than the susceptible DBM1Ac-S larvae, but the chymotrypsin activity was similar between these two *P. xylostella* strains (*p* < 0.05; Duncan’s tests; n = 3) (Figure 1A). Subsequently, the incubation of Cry1Ac protoxin with midgut protease extracts from susceptible DBM1Ac-S or resistant SZ-R larvae were compared (Figure 1B). After 1-h incubation, a strong single band of about 65 kDa was produced in DBM1Ac-S gut extracts, corresponding to the processed Cry1Ac protein, having a similar band size as produced by the control treatment with bovine trypsin. In contrast, two bands were observed in the activation produced by SZ-R gut extracts: one similar to that produced by the control bovine trypsin or DBM1Ac-S and the other of higher molecular weight. These results confirmed that the potential altered *PxTryp_SP* gene expression might be involved in Bt Cry1Ac resistance in SZ-R strain via decreasing midgut protease activities and protoxin activation.

### 2.2. Bioassay Analyses of Cry1Ac Protoxin and Activated Toxin

To further validate the influence of altered *PxTryp_SP* gene expression on the resistance to Cry1Ac toxin in SZ-R strain, bioassays with Cry1Ac (protoxin and activated toxin) were further conducted. Bioassays revealed that the resistance levels of Cry1Ac protoxin or activated toxin by trypsin were different in the SZ-R strain (Table 1). The resistance ratios (RR) were 662 for Cry1Ac protoxin and 422 for the activated Cry1Ac toxin in the SZ-R strain. The LC_50_ values of activated toxin and protoxin showed slightly but significant differences since the LC_50_ value of activated toxin was less than two fold lower to that of Cry1Ac protoxin (Table 1). These data correlated with the partial activation of Cry1Ac protoxin as shown above (Figure 1B). Furthermore, to verify whether Cry1Ac protoxin is about half as potent as the activated toxin due to their different molecular weights (130 vs. 65 kDa), we estimated the potency of Cry1Ac protoxin relative to activated toxin as reported before [33]. The potency ratios (PR) were 0.83 for the DBM1Ac-S strain and 0.53 for the SZ-R strain, which did not differ significantly from the predicted value of 0.50 (one sample *t*-test, df = 1, t = 3.25, *p* = 0.19), implying that the Cry1Ac protoxin was no more effective than the activated toxin in both strains analyzed (Table 1).

### 2.3. Cloning, Characterization, and Phylogenetic Analyses of the PxTryp_SPc1 Gene

During the previous characterization of differentially altered genes in SZ-R, we identified that a *PxTryp_SPc1* gene was possibly down-regulated [31,32]. Thus, we further explored this gene in *P. xylostella*. Based on the unigene sequences from the midgut transcriptome database of *P. xylostella* [32], the full-length cDNA sequence of *PxTryp_SPc1* gene (GenBank accession no. MN422356) was cloned from fourth-instar *P. xylostella* larval midgut tissue using specific primers (Appendix A). The cDNA sequence of the *PxTryp_SPc1* gene (799 bp) contains an ORF of 768 nucleotides encoding 222 amino acid residues. The genomic DNA (gDNA) sequence of this gene can be found in the *P. xylostella* genome (DBM-DB: http://iae.fafu.edu.cn/DBM, Gene ID: Px016056). The genomic analysis revealed that it contains four exons (Figure 2A). The amino acid sequence of the *PxTryp_SPc1* showed structural features characteristic of members of the trypsin family, as three catalytic residues His (^70^H), Ser (^116^S), and Asp (^211^D) (Figure 2B).

The PxTryp_SPc1 protein shares sequence identity from 17% to 52% with other insect trypsin orthologs, as revealed by the BLASTp homology search of the GenBank database (Appendix A). Moreover, phylogenetic analysis of different insect trypsin orthologs showed that trypsin proteins from different insect orders are clustered in independent branches and are evolutionarily conserved (Figure 2C). Additionally, the phylogenetic tree revealed close relationship among trypsin proteins from Lepidoptera and PxTryp_SPc1, which indicated that these trypsin proteins are homologous. Moreover, those trypsin proteins that were reported to be related to Bt resistance were not identified as PxTryp_SPc1 orthologs and were not included in this phylogenetic tree.

### 2.4. Tissue Expression Profiles of the PxTryp_SPc1 Gene

Expression analysis of the *PxTryp_SPc1* gene by qPCR in the different tissues of the fourth-instar larvae indicated that it was specifically expressed in the midgut (MG) tissue, in contrast to its expression in the head, integument, testis, and Malpighian tubules (*p* < 0.05; Duncan’s test; n = 3) (Figure 3A). Moreover, expression analysis of *PxTryp_SPc1* gene in various developmental stages showed that its expression levels gradually raised from egg (EG) into the larval stages and reached the highest peak in the fourth-instar larvae (L4), while it showed low expression in pre-pupae, pupae, female, and male adults (Figure 3B). The expression of *PxTryp_SPc1* gene was high in midgut and larval stages of *P. xylostella*.

### 2.5. The Expression of PxTryp_SPc1 Gene in Susceptible DBM1Ac-S and Resistant SZ-R Strains

Expression difference of the *PxTryp_SPc1* gene by qPCR was compared in the resistant SZ-R and susceptible DBM1Ac-S strains (Figure 4). In general, the transcript levels of *PxTryp_SPc1* showed a significantly reduced expression (about 2.8-fold down) in the SZ-R strain compared to the DBM1Ac-S strain (*p* < 0.05; Duncan’s test; n = 3). Furthermore, treatment of third-instar SZ-R larvae with a high concentration of Cry1Ac protoxin (2000 mg/L), showed that the transcript level of *PxTryp_SPc1* gene was further down-regulated (*p* < 0.05; Duncan’s test; n = 3), showing a ratio of ~5.1-fold down compared to the DBM1Ac-S strain (Figure 4).

### 2.6. Linkage between Decreased PxTryp_SPc1 Gene Expression and Cry1Ac Resistance in SZ-R Strain

To determine the genetic linkage of decreased *PxTryp_SPc1* expression with Cry1Ac resistance in the SZ-R strain, a single-pair cross between a male SZ-R larva and a female DBM1Ac-S larva was performed to obtain F1 progeny. Subsequently, backcross family a or b generated from reciprocal crosses between SZ-R moths and F1 progeny were selected and fed on cabbage leaves without or with a diagnostic dose of Cry1Ac protoxin (20 mg/L), and the midgut samples from fourth-instar *P. xylostella* larvae were subjected to qPCR analysis. The qPCR results indicated that *PxTryp_SPc1* gene expression levels in individual fourth-instar larval midguts from F1 generation resemble those in their susceptible DBM1Ac-S strain (Figure 5), implying that the resistance trait in SZ-R is recessive. Nevertheless, the expression levels of *PxTryp_SPc1* in midgut tissue from two backcross families (backcross a and b) showed two different groups; one displayed notable decreased expression levels of *PxTryp_SPc1* (< ~2.8-fold), but another group demonstrated similar expression levels to those of larvae midgut tissue from the original susceptible DBM1Ac-S strain or the F1 generation from the DBM1Ac-S and SZ-R strains cross (Figure 5). The ratios between the two families of individuals were found to be 10:8 (backcross a) and 9:9 (backcross b), following the calculated random assortment ratio 1:1 basically (*p* > 0.1 or *p* = 1.0; χ^2^ test). On the contrary, all of the survivals from Cry1Ac exposure in the two backcross families showed decreased expression levels of *PxTryp_SPc1* (<~2.8-fold) compared to larvae of the DBM1Ac-S strain or the F1 progenies, testifying a co-segregation (linkage) with resistance to Cry1Ac in SZ-R (*p* < 0.05, χ^2^ test) (Figure 5). Thus, the decreased expression level of the *PxTryp_SPc1* gene was tightly linked to Cry1Ac resistance in the *P. xylostella* SZ-R strain.

### 2.7. RNAi-Mediated Functional Assay of the PxTryp_SPc1 Gene

The *PxTryp_SPc1* gene expression was silenced by microinjection of *P. xylostella* susceptible larvae with *PxTryp_SPc1* dsRNA to determine the potential role of the *PxTryp_SPc1* gene in Cry1Ac resistance. The expression levels were statistically reduced after 24 h post dsRNA injection, and the reduction effect lasted almost 96 h. In contrast, controls treated with buffer or dsEGFP, did not show any silencing effect on *PxTryp_SPc1* expression (Figure 6A). The subsequent bioassays revealed that silencing of *PxTryp_SPc1* gene reduced larval susceptibility to Cry1Ac protoxin at 1 mg/L (the LC_50_ value) or at 2 mg/L (the LC_90_ value) after 48 h post-injection compared to control larvae injected with buffer or dsEGFP (Figure 6B), suggesting that the reduced expression of *PxTryp_SPc1* gene correlated with higher tolerance of *P. xylostella* to Cry1Ac. We also determined the effect of *PxTryp_SPc1* gene silencing on Cry1Ac protoxin activation using larval midgut extracts from different RNAi treatments. After 1 h of Cry1Ac incubation from larvae injected with nonspecific dsEGFP and buffer-only, a single band of ~65 kDa was observed (Figure 6C, lanes 4 and 6). In contrast, larvae treated with dsPxTryp_SPc1 showed the two bands of Cry1Ac as previously observed with midgut extract from the SZ-R strain (Figure 6C, lane 5 compared to Figure 6C lane 5).

## 3. Discussion

The role and function of the insect proteases present in midgut tissue in mediating Bt resistance has been analyzed in different *P. xylostella* resistant strains. It was shown before that decreased activation of protoxin to toxin could be a major Bt resistance mechanism in the Cry1Ac-resistant Cry1Ac-SEL strain of *P. xylostella* [35] and a Bt resistance strain of *P. xylostella* [36], but the specific midgut protease gene involved was not identified. Here, we show that the reduced expression of the *PxTryp_SPc1* trypsin gene contributes to Cry1Ac resistance in the Cry1Ac-resistant SZ-R strain. The activities of caseinolytic and trypsin proteases in midgut extracts significantly decreased in contrast to the susceptible strain, suggesting that reduced trypsin protease activity is associated with a resistant phenotype of the SZ-R strain. Based on previously published transcriptome, RNA-Seq, and proteomics-based studies [31,32,37], we identified a new trypsin gene, *PxTryp_SPc1* (GenBank accession no. MN422356, DBM-DB gene ID: Px016056), which is mainly expressed in the midgut and that is down-regulated in the SZ-R resistant strain. Previous identified trypsin proteins involved in Bt resistance in different lepidopteran insects including OnT23 (~50%, GenBank accession no. AAR98919), SfT6 (~43%, GenBank accession no. ACR25157), HaSP2 (~33%, GenBank accession no. ABP96915), and HaTryR (~28%, GenBank accession no. AHL46496) have been reported to be associated with Bt resistance in *O. nubilalis* [14], *S. frugiperda* [15], and *H. armigera* [16,17] (the percentage in parentheses means the identity between the amino acid sequences of these trypsins and PxTryp_SPc1). Although the protein identity between PxTryp_SPc1 and OnT23 or SfT6 is as high as 50% and 43%, respectively, another *P. xylostella* trypsin that shared 57% protein similarity with PxTryp_SPc1 was identified in the *P. xylostella* genome database (DBM-DB gene ID: Px015403). This *P. xylostella* trypsin has higher protein identity (55% and 63%, respectively) with OnT23 or SfT6, indicating that PxTryp_SPc1 and OnT23 or SfT6 are actually not orthologs. Some of the orthologs of PxTryp_SPc1 in other lepidopteran insects are shown in the phylogenetic tree constructed in this study (Figure 2C). The role of other orthologs from different lepidopteran insects in Cry toxin activation still remains to be identified. These results indicated that PxTryp_SPc1 is a novel trypsin member related to Bt resistance, which enriches the Bt-responsive midgut trypsin gene repertoire in insects.

The most common mechanism of high resistance levels to Bt Cry toxins in lepidopteran insects is related to reduced toxin binding to midgut receptors [7,38], while an altered protease expression mechanism has been associated with low or moderate resistance in lepidopteran insects [39]. Indeed, altered processing of Cry1Ac protoxin by midgut proteases is not related to high-level field-evolved Cry1Ac resistance in the *P. xylostella* NIL-R strain [30]. The laboratory-selected strain SZ-R shows moderate Cry1Ac resistance to Cry1Ac activated toxin or protoxin (Table 1). We reported that the *PxCAD* and *PxABCH1* genes are not associated with Cry1Ac resistance in the SZ-R strain [24], but differential expression of the *PxmALP*, *PxABCB1*, *PxABCC1*, *PxABCC2*, *PxABCC3*, and *PxABCG1* genes was shown to be associated with Cry1Ac resistance in the SZ-R strain [27,28,29], suggesting that reduction in toxin binding is associated with Cry1Ac resistance in the SZ-R strain. Thus, the resistance mechanism related to reduced expression of the trypsin gene *PxTryp_SPc1* identified in this study is an additional mechanism in this *P. xylostella* strain. The reduced expression of the *PxTryp_SPc1* gene correlated well with altered Cry1Ac protoxin activation, suggesting that incomplete activation of protoxin is conducive to developing the resistant phenotype (Figure 1B). However, our data showed that protoxin activation was not completely blocked in the SZ-R strain since treatment of Cry1Ac protoxin with midgut juice from the resistant population resulted in two protein bands; one correlated with the 65 kDa Cry1Ac activated toxin, implying that other midgut proteases may still participate in the activation of Cry1Ac protoxin in the resistant SZ-R strain. These data correlated with the toxicity bioassays revealed that SZ-R was only two-fold more susceptible to the Cry1Ac activated toxin compared with the protoxin (Table 1). Nevertheless, RNAi analysis showed that silencing *PxTryp_SPc1* gene did reduce the larval susceptibility to Cry1Ac toxin supporting that this protein contributes to the Cry1Ac resistance phenotype of SZ-R strain (Figure 6). Moreover, reduced expression of *PxTryp_SPc1* was linked to Cry1Ac resistance in SZ-R strain (Figure 5).

The regulation mechanism involved in the reduced expression of *PxTryp_SPc1* in the SZ-R strain still remains to be determined. Interestingly, down-regulation of the trypsin gene *HaTryR* in *H. armigera* is caused by a promoter sequence mutation mediated *cis*-regulatory mechanism [17]. Our previous studies demonstrated that different expression of the *PxmALP*, *PxABCB1*, *PxABCC1–3*, and *PxABCG1* genes can be modulated by the MAPK signaling pathway [27,29]. Thus, whether reduced expression of *PxTryp_SPc1* in SZ-R strain is conferred by the promoter mutation-induced *cis*- or MAPK-induced *trans*-regulatory mechanism warrants further study. In addition, considering that the *PxABCC2* and *PxABCC3* genes were successfully knocked out by a novel CRISPR/Cas9 genome editing tool confirming their involvement in Bt Cry1Ac resistance [40], we will further utilize the CRISPR/Cas9 system to edit the *PxTryp_SPc1* gene to offer in vivo reverse genetic evidence of its involvement in Cry1Ac resistance, which thus could help us in the future to determine what is the function of *PxTryp_SPc1* gene during Bt Cry1Ac toxin activation processing in *P. xylostella*. Moreover, the *P. xylostella* genome contains many different trypsin genes [41]; some other trypsin genes displaying marked differential expression levels between Cry1Ac-susceptible and -resistant *P. xylostella* strains have also been found previously by transcriptome, RNA-Seq, and proteomics analyses, and further investigations of their potential functions in Bt Cry1Ac resistance in *P. xylostella* are also needed.

Overall, our data confirmed that down-regulation of a novel trypsin gene (*PxTryp_SPc1*) is associated with Cry1Ac resistance in the SZ-R strain of *P. xylostella*. This study is helpful to elucidate the complex causes of Bt Cry1Ac resistance in *P. xylostella*. The deeper understanding that we have of these mechanisms, the stronger and better strategies we will be able to propose to cope with the evolution of insect resistance to Bt toxins.

## 4. Materials and Methods

### 4.1. Insect Strains

The *P. xylostella* susceptible DBM1Ac-S and resistant SZ-R strains that were used in this study were previously described [24,28]. The SZ-R strain was originated from field collected moths at Shenzhen in China (2003), and it was constructed by constant selection with a concentration of Cry1Ac protoxin that generally kills 50%–70% of larvae in the laboratory for more than 200 generations. Both *P. xylostella* strains were reared on Chinese cabbage, JingFeng No. 1 (*Brassica oleracea* var. *capitata*), at 65% RH, 25 °C, with a photoperiod of 16 h light:8 h dark, and adults were fed with a 10% sucrose solution.

### 4.2. Midgut Protease Activity Assays

The caseinolytic protease was measured at 28 °C using the substrate azocasein (Sigma, St. Louis, MO, USA), as previously reported [42]. In brief, midgut extracts (20 μL) were mixed with 1% azocasein in 50 mM NaHCO_3_-Na_2_CO_3_ buffer (150 μL) and incubated for 2 h at 28 °C. Then 10% trichloroacetic acid (TCA) (170 μL) was used to stop the reaction. The solution was incubated at 25 °C for 1 h and centrifuged for 15 min at 16,000 × *g* at room temperature to remove the debris. Then, 1 M NaOH (340 μL) was mixed and the optical density (OD) of collected supernatant was measured at 450 nm in a SpectraMaxM2*^e^* microplate reader (Molecular Devices, Sunnyvale, CA, USA).

Chymotrypsin and trypsin activities were detected using 1 mM Nα-benzoyl-L-arginine-*p*-nitroanilide (BApNA, Sigma) and 1 mM *N*-succinyl-Ala-Ala-Pro-phenylalanine *p*-nitroanilide (SAApFpNA, Sigma) as respective specific substrates. For chymotrypsin activity determination, 5 μL midgut extract was mixed with 3 mL of 1 mM SAApFpNA in 50 mM NaHCO_3_-Na_2_CO_3_ buffer. For trypsin activity examination, 10 μL midgut extract was mixed with 3 mL of 1 mM BApNA in 50 mM NaHCO_3_-Na_2_CO_3_ buffer. The peptidolytic reaction was tested immediately by recording continuously the optic density (OD) value at 405 nm every 15 s at 28 °C for 30 min. The enzyme activities are exhibited as relative activities of the DBM1Ac-S midgut extract protease activities, which were considered as 100%. Biological assay was performed in triplicate and four technical repetitions each were used to confirm the protease activities. For analysis of statistical differences among samples, one-way ANOVA with Duncan’s tests (*p* < 0.05) was used.

### 4.3. Bioassays

The Cry1Ac protoxin and trypsin-activated toxin were obtained as previously described [30]. The Cry1Ac toxin was finally dissolved in 50 mM Na_2_CO_3_ (pH 9.6) and stored at −20 °C. The respective toxicity of the Cry1Ac protoxin and trypsin-activated toxin was determined by 72-h leaf-dip bioassays using a total of 280 third-instar *P. xylostella* larvae per bioassay as described before [27,28]. In short, ten larvae that were exposed to seven different concentrations of Cry1Ac toxin in each group, and four repeats were performed for all bioassays. The control mortality did not exceed 5%. We used the POLO Plus 2.0 software (LeOra Software, Berkeley, CA, USA) to calculate the LC_50_ values (median lethal concentrations killing 50% of the tested larvae) and 95% CL (95% confidence limits of the LC_50_) values by Probit analysis.

### 4.4. RNA Extraction and cDNA Synthesis

The methods of RNA extraction and cDNA synthesis from *P. xylostella* were previously described [27]. The midgut samples were extracted in TRIzol Reagent (Invitrogen, Carlsbad, CA, USA); then the concentration of RNA was quantified by a NanoDrop 2000c spectrophotometer (Thermo Fisher Scientific Inc., Waltham, MA, USA). PrimeScript II 1st strand cDNA Synthesis Kit (TaKaRa, Dalian, China) was used to synthetize first-strand DNA. For qPCR analysis, 1 μg total RNA was used to perform the first-strand cDNA with the PrimeScript RT kit (containing gDNA Eraser, Perfect Real Time) (TaKaRa, Dalian, China) following the manufacturer’s instructions. The synthesized cDNA was immediately used or stored at −20 °C until used.

### 4.5. Gene Identification and Cloning

The candidate cDNA sequence of *PxTryp_SPc1* gene was identified in our previous midgut transcriptome database of *P. xylostella* [32] and was further in silico corrected by the *P. xylostella* genome database (DBM-DB: http://iae.fafu.edu.cn/DBM, Gene ID: Px016056); then the specific primers were designed (Appendix A) and were used in subsequent PCR amplification assays. The full-length cDNA sequence *of PxTryp_SPc1* gene was finally obtained and deposited in the GenBank database (accession no. MN422356).

As described previously [27], the PCR reaction (25 μL total volume) contained 0.2 μL LA Taq HS polymerase (TaKaRa, Dalian, China) in an C1000 Thermal Cycler PCR system (BioRad, Philadelphia, PA, USA) for 35 cycles using LA Taq polymerase (TaKaRa, Dalian, China). A gel extraction kit (CWBIO, Beijing, China) was used for purification of the PCR products of *PxTryp_SPc1*, which were further cloned into the pEASY-T1 vector (TransGen, Beijing, China). For gene sequencing, *Escherichia coli* TOP10 competent cells (TransGen, Beijing, China) were transformed with candidate plasmids.

### 4.6. Gene Sequence Analysis

DANMAN 8.0 (Lynnon BioSoft, San Ramon, CA, USA) software was used for gene sequence assembly, exon-intron analysis, and multiple sequence alignment. The open reading frame (ORF) was identified by the ORF finder tool at the NCBI (https://www.ncbi.nlm.nih.gov/orffinder/), and predicted amino acid sequences were achieved by ExPASy online tool to translate (https://web.expasy.org/translate/). The BLAST tool at the GenBank database (https://blast.ncbi.nlm.nih.gov) was used for the sequence-similarity analyses. The protein-specific motifs and active sites were found and annotated at the GenBank database (https://www.ncbi.nlm.nih.gov/). The signal peptide was predicted by SignalP-5.0 Server online (http://www.cbs.dtu.dk/services/SignalP/).

### 4.7. Phylogenetic Tree Construction

To verify the classification of the *PxTryp_SPc1* gene, phylogenetic analysis of the PxTryp_SPc1 protein was done by using the full-length amino acid sequences of its orthologs from other insects. MEGA-X software (https://www.megasoftware.net/) with ClustalW algorithm was used to construct the phylogenetic tree. An unrooted neighbor-joining (NJ) phylogenetic tree was done choosing the “p-distance” as the amino acid substitution model; the bootstrap value was determined from 1000 replicates.

### 4.8. Sample Preparation

Samples from different developmental stages were collected, and different tissues were also dissected from the fourth-instar DBM1Ac-S larvae to characterize the spatio-temporal expression patterns of the *PxTryp_SPc1* gene. Moreover, in order to resolve whether the expression level of *PxTryp_SPc1* was related to Cry1Ac resistance, third-instar SZ-R larvae were treated with a high concentration of Cry1Ac protoxin (2000 mg/L). After the midguts from the survivors were dissected, the extraction of total RNA and cDNA was synthesized as mentioned above. Data were obtained from three biological replications performed in all samples.

### 4.9. Gene Expression Analysis

Gene expression differences were determined by real-time quantitative PCR (qPCR) as described before with slight modification [27,28]. Briefly, Primer Premier 5.0 (PREMIER Biosoft international, Palo Alto, CA, USA) was used for defining specific *PxTryp_SPc1* gene primers (Appendix A). PCR reactions (20 μL) contained 7.4 μL RNase-Free ddH_2_O, 10 μL of 2 × FastFire qPCR PreMix Plus (TIANGEN, Beijing, China), 5 μM of each specific primer, 1 μL of first-strand cDNA template, and 0.4 μL 50 × ROX Reference Dye (TIANGEN, Beijing, China). The running program consisted of a denaturation at 95 °C for 10 min followed by 40 denaturalized cycles at 95 °C for 15 s, annealing at 57 °C for 30 s, and extension at 72 °C for 30 s. All reactions were performed in an Applied Biosystems QuantStudio 3 Real-Time PCR System (Applied Biosystems, Forster City, CA, USA). As an internal control for relative quantification, the *ribosomal protein L32* (*RPL32*) gene (GenBank accession no. AB180441) was used in qPCR data analysis. Three biological repetitions and four technical repetitions were conducted for each sample. To define the statistically differences, one-way ANOVAs with Duncan’s test (overall significance level *p* < 0.05) were used.

### 4.10. Linkage Analysis

Genetic linkage analysis was performed as previously described [27,28]. F1 progeny was generated by a single-pair mating between a SZ-R male and a DBM1Ac-S female. A diagnostic Cry1Ac protoxin diagnostic dose (20 mg/L) killed all the F1 (heterozygous) larvae was determined in a toxicity bioassay. Reciprocal crosses between SZ-R moths and F1 progeny were made to generate backcross family a and b. Forty larvae from each backcross families of progeny were fed on cabbage (non-Cry1Ac-selected) or cabbage with 20 mg/L of Cry1Ac protoxin (Cry1Ac-selected), and midguts tissues from the survived fourth-instar *P. xylostella* larvae were dissected for qPCR analysis as mentioned above.

### 4.11. RNA Interference (RNAi)

To determine the impact of *PxTryp_SPc1* gene expression in *P. xylostella* resistance to Cry1Ac, RNAi-mediated down expression of *PxTryp_SPc1* gene was performed. Early third-instar *P. xylostella* larvae were microinjected with specific dsRNA targeting *PxTryp_SPc1* gene (dsPxTryp_SPc1), as described previously [24]. Briefly, Primer Premier 5.0 (PREMIER Biosoft international, Palo Alto, CA, USA) was used to design the dsRNA primers containing the T7 promoter on the 5′ end targeting gene-specific region of *PxTryp_SPc1* (GenBank accession no. MN422356) or *EGFP* gene (GenBank accession no. KC896843) (Appendix A). To further validate the specificity of these dsRNAs, BLASTn searches of the GenBank database (https://www.ncbi.nlm.nih.gov/) and the *P. xylostella* genome database (DBM-DB: http://iae.fafu.edu.cn/DBM) were performed showing no unspecific hit diminishing potential off-target effects. The amplicons (389 bp for dsPxTryp_SPc1 and 469 bp for dsEGFP) were used as templates for in vitro transcription reactions to produce dsRNAs using the T7 RiboMAX Express RNAi System (Promega, Madison, WI, USA). The synthesized dsRNA was dissolved in 10 mM Tris–HCl (pH 7.0), and 1 mM EDTA was used as injection buffer and mixed with Metafectene PRO transfection reagent (Biontex, Planegg, Germany). A total of 300 ng (70 nL) dsEGFP or dsPxTryp_SPc1 were injected into the hemocoel of DBM1Ac-S larvae, resulting in less than 20% larval mortality determined after 5 days. Finally, to determine the silencing efficiency at 48 h post-injection, midgut tissue was dissected from injected larvae. The control group was injected with equal volumes of buffer alone. At least 30 larvae were analyzed for each treatment, and three replicate experiments were conducted. The bioassay data were processed as mentioned above. Statistically significant differences between qPCR and bioassay analyses were determined by one-way ANOVAs with Duncan’s tests (overall significance level *p* < 0.05).

## Figures and Tables

**Figure 1 toxins-12-00076-f001:**
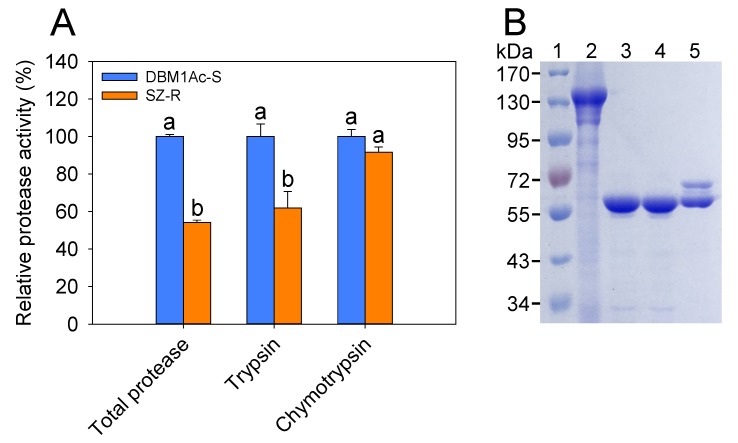
Midgut protease activities (caseinolytic proteases, trypsin, and chymotrypsin) and the activation differences between the *P. xylostella* DBM1Ac-S and SZ-R strains. (**A**). Protease activities were calculated relative to the activities shown by the susceptible DBM1Ac-S strain (100%). Different letters stand for statistically significant differences within the three replicates and four technical repeats (*p* < 0.05; Duncan’s test; n = 3). (**B**) Activation of Cry1Ac protoxin with protease midgut extracts from control (DBM1Ac-S) or resistant (SZ-R) strains. Lane 1: protein maker; Lane 2: Cry1Ac protoxin; Lane 3: Cry1Ac incubated with bovine trypsin (positive control); Lane 4: Cry1Ac incubated with protease midgut extracts from DBM1Ac-S strain; Lane 5: Cry1Ac incubated with midgut extracts from SZ-R strain.

**Figure 2 toxins-12-00076-f002:**
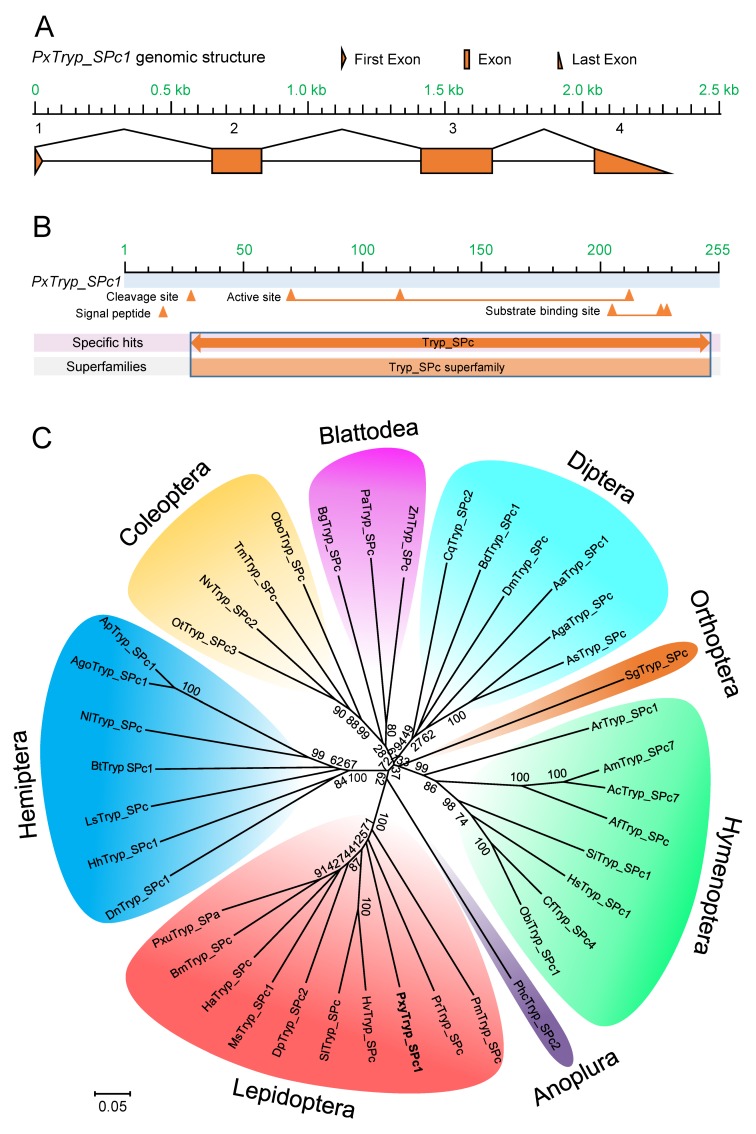
Structural and phylogenetic relationship analyses of *PxTryp_SPc1* gene. (**A**) Genomic structure of *PxTryp_SPc1* gene. Orange boxes represent exons, and the spaces between two boxes represent introns. The figure is drawn to scale. (**B**) Conserved domain annotation obtained from NCBI annotation of the PxTryp_SPc1 protein sequence. The protein sequence was considered as a characteristic member of the trypsin family. The location of the signal peptide, cleavage site, active sites, and substrate binding sites are indicated by orange triangles. (**C**) Phylogenetic analysis of the PxTryp_SPc1 protein and its orthologs in diverse insects by the neighbor-joining (NJ) method. The unrooted phylogenetic tree was constructed by ClustalW alignment of amino acid sequences in MEGA-X. The bootstrap values with 1000 replications are shown on branches. The amino acid sequences of these trypsins were retrieved from the GenBank database (GenBank accession numbers are listed below). The scale bar shows the evolutionary distances. Abbreviations: 1. Lepidoptera (Pm (*Papilio machaon*, KPJ14943); Pr (*Pieris rapae*, XP_022118678); Pxy (*Plutella xylostella*, MN422356); Hv (*Heliothis virescens*, AFO68329); Sl (*Spodoptera litura*, XP_022815738); Dp (*Danaus plexippus*, OWR45697); Ms (*Manduca sexta*, CAM84320); Ha (*Helicoverpa armigera*, ABU98624); Bm (*Bombyx mori*, XP_004923288); Pxu (*Papilio xuthus*, KPJ03461)); 2. Hemiptera (Dn (*Diuraphis noxia*, XP_015367971); Hh (*Halyomorpha halys*, XP_024219146); Ls (*Laodelphax striatellus*, RZF38227); Bt (*Bemisia tabaci*, XP_018896298); Nl (*Nilaparvata lugens*, XP_022184709)); Ago (*Aphis gossypii*, XP_027850262); Ap (*Acyrthosiphon pisum*, XP_001943273)); 3. Coleoptera (Obo (*Oryctes borbonicus*, KRT83696); Tm (*Tenebrio molitor*, AFB81537); Nv (*Nicrophorus vespilloides*, XP_017773892); Ot (*Onthophagus taurus*, XP_022900611)); 4. Blattodea (Bg, (*Blattella germanica*, AAZ78212); Pa (*Periplaneta americana*, AIA09342); Zn (*Zootermopsis nevadensis*, XP_021914447)); 5. Diptera (As (*Anopheles sinensis*, KFB42846); Aga (*Anopheles gambiae*, XP_317171.2); Aa (*Aedes aegypti*, XP_001657786); Bd (*Bactrocera dorsalis*, XP_011214086); Cq (*Culex quinquefasciatus*, XP_001847028); Dm (*Drosophila melanogaster*, NP_001285772)); 6. Orthoptera (Sg, (*Schistocerca gregaria*, CAA70820)); 7. Hymenoptera (Ar (*Athalia rosae*, XP_020711972); Am (*Apis mellifera*, XP_623564); Ac (*Apis cerana*, XP_016922703); Af (*Apis florea*, XP_012344846); Si (*Solenopsis invicta*, XP_011166798); Hs (*Harpegnathos saltator*, EFN81462); Cf (*Camponotus floridanus*, XP_011266670); Ob (*Ooceraea biroi*, XP_011336015)); 8. Anoplura (Phc (*Pediculus humanus corporis*, AAV48634)).

**Figure 3 toxins-12-00076-f003:**
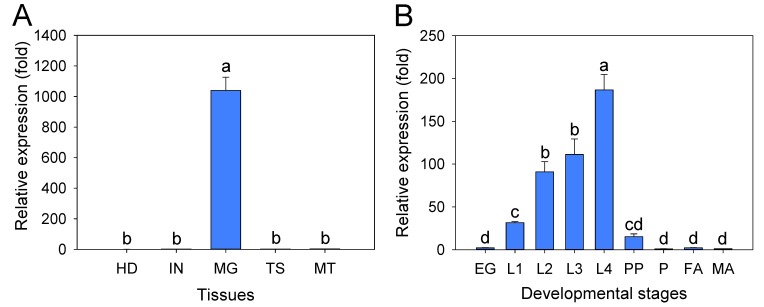
qPCR expression profile of the *P. xylostella PxTryp_SPc1* gene in different tissues and developmental stages. (**A**) Relative expression levels of *PxTryp_SPc1* in different tissues including head (HD), integument (IN), midgut (MG), testis (TS), and Malpighian tubules (MT) of fourth-instar larvae. (**B**) Expression profile of *PxTryp_SPc1* in different developmental stages: eggs (EG), first-instar larvae (L1), second-instar larvae (L2), third-instar larvae (L3), fourth-instar larvae (L4), prepupae (PP), pupae (P), male adults (MA), and female adults (FA). *RPL32* gene expression was used as the internal reference gene to normalize and calculate the gene expression levels. Expression level was calculated according to the value of the lowest expression identified (Tissues: HD; developmental stages: P), which was given an arbitrary value of 1. The means and the corresponding standard errors are shown. Different letters stand for statistically significant differences within the three replicates and four technical repeats (*p* < 0.05; Duncan’s test; n = 3).

**Figure 4 toxins-12-00076-f004:**
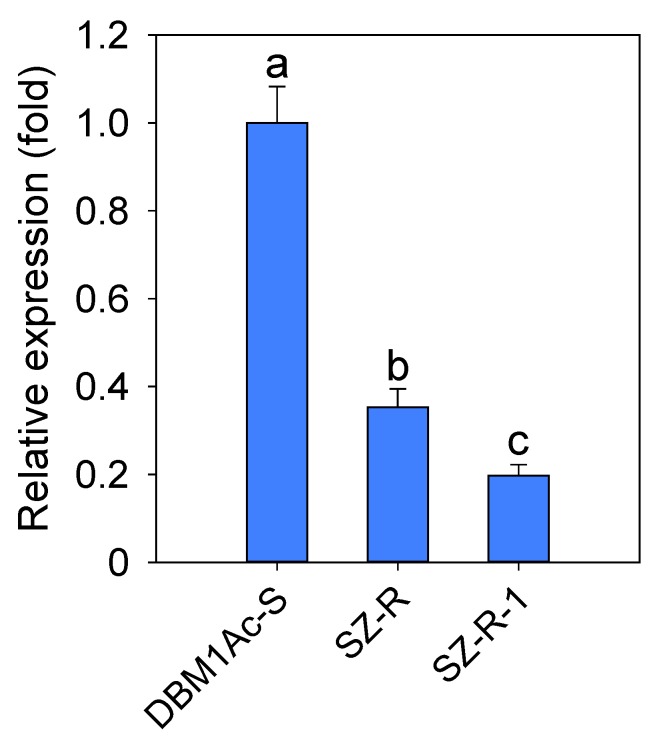
The expression differences of *PxTryp_SPc1* gene between susceptible and resistant strains. Expression levels of *PxTryp_SPc1* by qPCR in fourth-instar larval midgut tissue from susceptible and resistant strains. Lane 1: DBM1Ac-S; Lane 2: SZ-R; Lane 3: SZ-R intoxicated with 2000 mg/L Cry1Ac protoxin. *RPL32* gene was considered as a reference gene to normalize and calculate the level of gene expression. The expression level was calculated based on the value of the highest expression (DBM1Ac-S, arbitrary value of 1). The means and standard errors are shown. Different letters represent statistically significant differences with three independent repeats and four technical replications (*p* < 0.05; Duncan’s test, n = 3).

**Figure 5 toxins-12-00076-f005:**
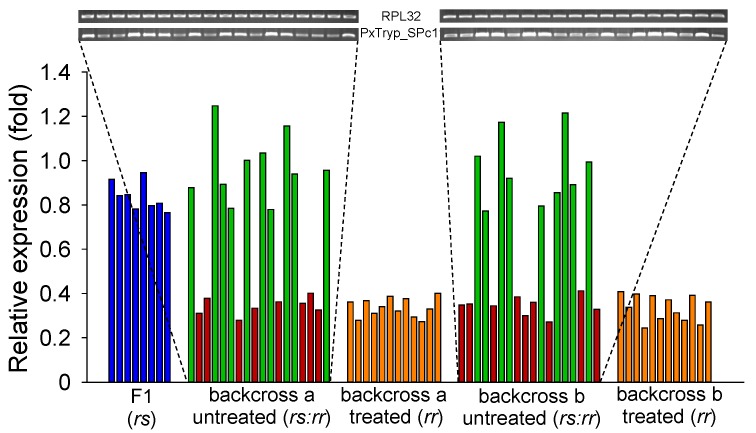
Genetic linkage analysis of the decreased *PxTryp_SPc1* expression level in the SZ-R strain of *P. xylostella* and resistance to Cry1Ac. The expression levels of *PxTryp_SPc1* in F1 larvae, Cry1Ac-treated backcross families (family a and b), and non-selected (untreated) are shown in relation to the levels in the DBM1Ac-S strain. Corresponding intensity of PCR bands for the *PxTryp_SPc1* and the reference *RPL32* gene are exhibited (Upper).

**Figure 6 toxins-12-00076-f006:**
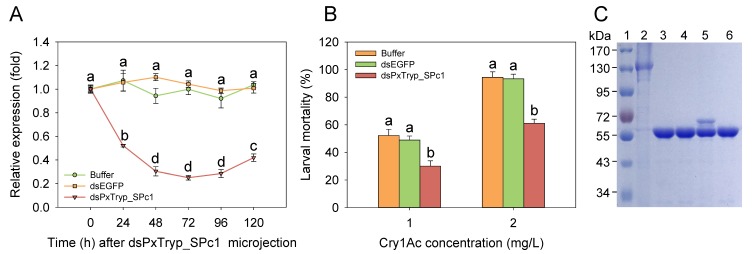
Influences on Cry1Ac toxicity of *PxTryp_SPc1* expression in larval midgut after RNAi silencing. (**A**) Impacts on injection of larvae with buffer, dsEGFP, or dsPxTryp_SPc1 on *PxTryp_SPc1* expression after 120 h RNAi silencing from *P. xylostella*. Different letters represent statistically significant differences within three repeats and four technical replications (*p* < 0.05; Duncan’s test, n = 3). (**B**) Mortality of *P. xylostella* larvae after treatment with two concentrations of Cry1Ac protoxin; larvae were injected with buffer, dsEGFP, or dsPxTryp_SPc1. Within each group, different letters denote statistically significant differences between treatments (*p* < 0.05; Duncan’s test, n = 3). (**C**) Activation analysis of Cry1Ac protoxin by *P. xylostella* midgut extracts from larvae injected with: dsEGFP (lane 4), dsPxTryp_SPc1 (lane 5), and buffer only (lane 6). Lane1: protein maker; Lane 2: Cry1Ac protoxin; Lane 3: Cry1Ac incubated with bovine trypsin (positive control).

**Table 1 toxins-12-00076-t001:** Bioassays of Cry1Ac protoxin and activated toxin in DBM1Ac-S and resistant SZ-R larvae.

Strains	Treatments	Slope (± SEM) ^a^	LC_50_ (95% FL) ^b^	RR ^c^	PR ^d^
DBM1Ac-S	Protoxin	2.008 (±0.232)	0.83 (0.64–1.06)	1.0	
SZ-R	Protoxin	1.815 (±0.252)	549.62 (411.60–797.83) *	662	
DBM1Ac-S	Activated toxin	2.156 (±0.252)	0.69 (0.54–0.87)	1.0	0.83
SZ-R	Activated toxin	1.561 (±0.200)	291.04 (216.61–408.95) *	422	0.53

^a^ Slope of the dose response-mortality. SEM stands for standard error of the mean. ^b^ LC_50_ (95% FL): Toxin concentration (mg/L) killing 50% of larvae and its 95% fiducial limits (lower-upper). ^c^ RR: Resistance ratio calculated by the ratio between the LC_50_ value of SZ-R by the LC_50_ of DBM1Ac-S. ^d^ PR: Potency was calculated as the ratio of LC_50_ value of activated toxin by the LC_50_ of protoxin as reported [33,34]. Potency values < 1 indicate the activated Cry1Ac toxin is more potent than protoxin, while potency values > 1 indicate that Cry1Ac protoxin is more potent than activated toxin. * Asterisks represent significantly different LC_50_ value by the conservative criterion of non-overlapped 95% CL value.

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
