# Peer review of "Reduced Expression of a Novel Midgut Trypsin Gene Involved in Protoxin Activation Correlates with Cry1Ac Resistance in a Laboratory-Selected Strain of Plutella xylostella (L.)"

_toxins, 2020, doi:10.3390/toxins12020076_

Round 1
Reviewer 1 Report
This paper shows that decreased expression of a novel trypsin-like protease is involved in Bt resistance in the Cry1Ac-resistant P. xylostella strain SZ-R. The authors have cloned and characterized this gene (PxTryp_SPc1) and demonstrated that down-regulation of PxTryp_SPc1 gene in the midgut is related to Cry1Ac resistance in the by combining genomic, biochemical and genetic tools. They first showed that the resistant strain that shows decreased PxTryp_SPc1 gene expression as compared to the susceptible strain, also exhibits decreased midgut protease activities and Cry1Ac protoxin activation. Bioassays using protoxin or activated toxin suggest that part of the resistance involve the activation process. Expression profiles of PxTryp_SPc1 across larval stages and tissues (qPCR) show it is expressed mainly in susceptible L4 midgut. Controlled crosses between resistant and susceptible individuals show that resistance is recessive and linked to PxTryp_SPc1 expression level. Further validation comes from RNAi-mediated functional assay of the PxTryp_SPc1 gene. The manuscript is well written, the methods and analyses are adequate, and the results are clearly presented; the discussion highlights the role of downregulation of this midgut trypsin in Bt-resistance although recognizing that other mechanisms are at play. Conclusions of this study provide a new insight into the Bt resistance mechanism that could give hints for the control of insect pests.
Author Response
We thank the reviewer for recognizing the relevance of the data presented and the positive comments.
Reviewer 2 Report
Dear Authors,
present an interesting study using TCry1Ac in a laboratory-selected strain of Plutella xylostella. They authors do a nice job of setting up the introduction and the rationale for using Bt in Plutella xylostella resistance. Overall, the flow of the paper is good and I think they have sufficient data and good scientific integrity. My specific comments is in pdf file.

Author Response
Point 1: The authors present an interesting study using Cry1Ac in a laboratory-selected strain of Plutella xylostella. They authors do a nice job of setting up the introduction and the rationale for using Bt in Plutella xylostella resistance. Overall, the flow of the paper is good and I think they have sufficient data and good scientific integrity. My specific comments is in pdf file.
Response 1: We thank the reviewer for the positive comments, and we think our responses and actions described below correctly answer all concerns raised by the reviewer.
Point 2: (Lines 50-52) Please add full names. Binomial name, Author (Order, Family)
Response 2: Corrected (Lines 50-54).
Point 3: (Line 356) What are laboratory conditions? This is too broad of a statement
Response 3: The SZ-R strain was constructed by constant selection in the laboratory for more than 200 generations with a concentration of Cry1Ac protoxin generally kill 50–70% larvae. We have revised the corresponding sections (Lines 353-354) to address the reviewer’s concern.
Point 4: how many individuals you tested? Describe the block of the experiment
Response 3: In each bioassay test, we analyzed a total of 280 third-instar P. xylostella larvae for Cry1Ac protoxin and trypsin-activated toxin, respectively. We have revised the corresponding section to describe it (Lines 380-381). In each experiment, ten larvae were tested with seven different concentrations of Cry1Ac toxin in each group, and four repeats were done.
Reviewer 3 Report
The manuscript describes the contribution of decreased trypsin-like activity in the resistance in one strain of Px. Although this contribution is small, compared with supposed decreased binding that the authors mention they found previously in the same strain, the results are sound, indicating that altered activation of the Cry1Ac toxin contributes to the overall resistance levels.
Minor comments:
Lines 155-6, and 194-5: I do not agree with the statement of "probably involved in insect Bt resistance" or "suggesting that the .... may play an important role in Bt Cry1Ac resistance in Px". It is correct to say that it is a new gene, but its function as a mechanism of resistance is only shown at the end of the manuscript, and the statement is only valid for those strains sharing this same mechanism, though the gene is present in all Px strains, susceptible or resistant. Simply delete those sentences.
Lines 204-5: The other way round: (Tissues: HD; developmental stages: P)
Paragraph 2.6: It is not clear how the crosses are done. Only after reading M&M, at the end; one can understand the experimental design. Please make it a little more clear here.
Line 271: "after 48 h RNAi" is not correct in panel A.
Author Response
Point 1: The manuscript describes the contribution of decreased trypsin-like activity in the resistance in one strain of Px. Although this contribution is small, compared with supposed decreased binding that the authors mention they found previously in the same strain, the results are sound, indicating that altered activation of the Cry1Ac toxin contributes to the overall resistance levels.
Response 1: We thank the reviewer for the comments.
Minor comments:
Point 2: Lines 155-6, and 194-5: I do not agree with the statement of "probably involved in insect Bt resistance" or "suggesting that the .... may play an important role in Bt Cry1Ac resistance in Px". It is correct to say that it is a new gene, but its function as a mechanism of resistance is only shown at the end of the manuscript, and the statement is only valid for those strains sharing this same mechanism, though the gene is present in all Px strains, susceptible or resistant. Simply delete those sentences.
Response 2: We have deleted those sentences as suggested by the reviewer.
Point 3: Lines 204-5: The other way round: (Tissues: HD; developmental stages: P)
Response 3: Revised accordingly (Lines 203-204).
Point 4: Paragraph 2.6: It is not clear how the crosses are done. Only after reading M&M, at the end; one can understand the experimental design. Please make it a little more clear here.
Response 4: Revised accordingly (Lines 227-230).
Point 5: Line 271: "after 48 h RNAi" is not correct in panel A.
Response 5: We have changed “48 h” into “120 h” (Line 268).